# Functional Analysis of Mannosyltransferase-Related Genes *UvALGs* in *Ustilaginoidea virens*

**DOI:** 10.3390/ijms26072979

**Published:** 2025-03-25

**Authors:** Shilong Wang, Yating Zhang, Lili Qu, Zengran Zhou, Hongyang Zhai, Songhong Wei, Yan Wang

**Affiliations:** 1Department of Plant Pathology, College of Plant Protection, Shenyang Agricultural University, Shenyang 110866, China; wshilong0720@163.com (S.W.); yatingz2022@163.com (Y.Z.); 2024200180@stu.syau.edu.cn (Z.Z.); 2022220506@stu.syau.edu.cn (H.Z.); 2Liaoning Provincial Plant Protection and Quarantine Station, Shenyang 110034, China; qll_lat@163.com

**Keywords:** *Ustilaginoidea virens*, pathogenicity, Agrobacterium tumefaciens-mediated transformation, target gene knockout

## Abstract

Rice false smut, caused by *Ustilaginoidea virens*, is one of the three major rice diseases in China. It not only seriously affects the rice yield and quality but also endangers human and animal health. Studying the pathogenic mechanism of *U. virens* has important theoretical significance and application value for clarifying the infection characteristics of the pathogen and cultivating disease-resistant varieties. Plant pathogenic fungi utilize secreted effectors to suppress plant immune responses, which can function in the apoplast or within host cells and are likely glycosylated. However, the posttranslational regulation of these effectors remains unexplored. Deletion of ΔUvALG led to the cessation of secondary infection hyphae growth and a notable decrease in virulence. We observed that ΔUvALG mutants triggered a significant increase in reactive species production within host cells, akin to ALG mutants, which plays a crucial role in halting the growth of infection hyphae in the mutants. ALG functions by sequestering chitin oligosaccharides to prevent their recognition by the rice chitin elicitor, thereby inhibiting the activation of innate immune responses, including reactive species production. Our findings reveal that ALG3 possesses three N-glycosylation sites, and the simultaneous Alg-mediated N-glycosylation of each site is essential for maintaining protein stability and chitin-binding activity, both of which are critical for its effector function. These outcomes underscore the necessity of the Alg-mediated N-glycosylation of ALG to evade host innate immunity.

## 1. Introduction

Rice false smut (RFS) is a fungal disease caused by *Ustilaginoidea virens* (Cooke) Takah or *Villosiclava virens* (Nakata) E. Tanaka et C. Tanaka, leading to the formation of greenish spore balls in panicles and spikelets [1,2,3]. The disease was initially identified in India in the 1870s (Cooke, M.C., 1878) and has since spread extensively to all primary rice-producing regions globally, encompassing over 40 countries [4,5]. Despite the availability of numerous disease-resistant rice varieties, the majority of cultivated strains remain vulnerable to false smut infection [3,5]. The causal agent, *Ustilaginoidea virens* (anamorph), belongs to the Ascomycota phylum and produces asexual chlamydospores, while its sexual stage, *Villosiclava virens* (teleomorph), falls under the Ascomycetes class and generates sexual ascospores. The initial stages of the disease are asymptomatic, with symptoms only becoming apparent when rice grains are replaced by spherical fungal mycelia [6]. By this point, disease management is no longer effective [1]. False smut spreads through spores, with *U. virens* sori predominantly found in the palea and lemma of rice [7]. During late autumn, lower temperatures can trigger sclerotial differentiation and formation, leading to an increase in sclerotia numbers [8]. Sclerotia and chlamydospores are pivotal in the fungus’s life cycle, contributing to primary infections in subsequent years [9]. False smut significantly impacts both the quantity and quality of rice [10], while the fungi’s toxic secondary metabolites pose risks to humans and livestock. It has become a prevalent disease in Chinese rice paddies [11,12,13]. Currently, chemical fungicides are commonly employed for disease prevention and control [14].

Proteins secreted, such as cell wall glycoproteins and effector proteins, are synthesized in the endoplasmic reticulum (ER) and commonly undergo N-glycosylation and/or O-glycosylation [15]. N-Linked glycosylation is a prevalent posttranslational modification that confers proteins with unique folding, quality control, sorting, degradation, and secretion properties, as well as a recognition marker [15,16]. This type of glycosylation has been extensively researched in higher eukaryotes. In mammals, N-linked glycosylation participates in various cellular processes including cell adhesion, molecular trafficking, receptor activation, signal transduction, endocytosis, and is crucial for human health and disease. In higher plants, enzymes responsible for protein N-glycosylation and N-glycosylated glycoproteins have been extensively characterized and are known to play essential roles in diverse aspects of development and physiology, such as salt tolerance and plant immunity.

N-Glycosylation of proteins in eukaryotic cells occurs in the following two distinct organelles: the endoplasmic reticulum (ER) and the Golgi apparatus [16]. The process initiates with N-glycan biosynthesis, commencing with the synthesis of a precursor, Man5-GlcNAc2, attached to dolicholphosphate on the cytoplasmic side of the ER [17,18]. Subsequently, this precursor is translocated to the luminal side of the ER, where four mannose (Man) and three glucose (Glc) residues are sequentially added by various glycosyltransferases. The resulting oligosaccharide, Glc3Man9GlcNAc2, is then collectively transferred by the oligosaccharyltransferase complex onto specific asparagine (Asn) residues of secreted proteins [19,20]. Following protein folding and oligomerization, the glycoproteins are transported to the Golgi complex for further N-linked glycan modifications [21]. Complex N-glycans are formed in mature glycoproteins through the addition of extra saccharide units in the Golgi apparatus [22].

The yeast Saccharomyces cerevisiae is commonly utilized as a model organism for studying eukaryotic N-glycan synthesis. Investigations involving mutants with impaired N-glycan biosynthesis have facilitated the identification and characterization of the asparagine-linked glycosylation (ALG) genes [19,23]. Mutants of ALG in yeast exhibit deficiencies in either donor substrate biosynthesis or lipid-linked oligosaccharide assembly [20,24], providing insights into the role of N-glycosylation. Deletion of certain ALG genes responsible for initial stages of N-glycosylation leads to severe phenotypes, including lethality and the secretion of proteins with inadequate glycosylation, highlighting the significance of N-linked glycosylation [20]. In Candida albicans, N-glycosylation is crucial for the structure and function of specific proteins involved in essential cellular processes, such as cell–cell interactions, adhesion, and response to host immune defenses [25,26,27]. Nevertheless, the involvement of N-glycosylation in the pathogenesis of filamentous fungi remains largely unexplored. Only a limited number of studies, specifically three reports on glycosylation-related mutants in *Ustilago maydis* and Mycosphaerella graminicola, have suggested a role for N-glycosylation in the pathogenicity of plant pathogenic fungi [28,29,30]. However, the precise mechanism underlying this phenomenon remains unelucidated in these investigations, leaving the importance of N-glycosylation in fungal pathogenesis unclear [31,32,33].

In this study, the gene function of three mannosyltransferase genes in Ustilaginoidea virens was preliminarily explored. The results are as follows:(1)Thirty-eight genes encoding mannosyltransferases in *Ustilaginoidea virens* were screened by gene annotation. The expression levels of these genes at different stages of infection of *Ustilaginoidea virens* and different days of growth of *Ustilaginoidea virens* on medium were analyzed. The significantly upregulated genes *Uv8b_04477*, *Uv8b_01893*, and *Uv8b_04469* encoding mannosyltransferase were screened and analyzed by bioinformatics. The protein domain and physicochemical properties of these genes, the location of coding genes on chromosomes, the prediction of protein function, and the construction of a phylogenetic tree were predicted and analyzed. It was found that these three proteins contained different domains, which were ALG3, GT15, and OCH1, respectively. The proteins are consistent with their corresponding homologous gene domains. The amino acid lengths of the three proteins are about 400 aa, which is relatively stable. This included two hydrophilic proteins and one water-transporting protein. All chromosome sequences of Ustilaginoidea virens were downloaded from NCBI. There are seven chromosomes in *Ustilaginoidea virens*. *UvALG32* is located on chromosome 2, and *UvALG3* and *UvALG15* are located on chromosome 3. Homologous alignment and phylogenetic analysis of *UvALG3*, *UvALG15*, and *UvALG32* with mannosyltransferases from *Fusarium graminearum* and *Saccharomyces cerevisiae* were performed. We found that *UvALG3* had the highest homology with *S. cerevisiae ALG3* (96%), *UvALG15* had the highest homology with *S. cerevisiae KTR1* (100%), and *UvALG32* had the highest homology with *Fusarium graminearum OCH1* (100%).(2)In order to further study the function of the mannosyltransferase-encoding genes *UvALG3*, *UvALG15*, and *UvALG32*, based on the vector PXEH, the gene knockout vectors PXEHA3, PXEHA15, and PXEHA32 were constructed by *Agrobacterium*-mediated genetic transformation. The knockout transformants were obtained by *Agrobacterium*-mediated transformation and verified by PCR experiments. Finally, the knockout mutants ΔUvALG3, ΔUvALG15, and ΔUvALG32 were selected for subsequent experiments. Based on the vector PXNPTII, *UvALG3*, *UvALG15*, and *UvALG32* complement vectors CPXEHA3, CPXEHA15, and CPXEHA32 were constructed. Using the same techniques and methods as gene knockout, the gene complement mutants CΔUvALG3, CΔUvALG15, and CΔUvALG32 were finally screened.(3)The biological characteristics such as the growth rate, sporulation, and pathogenicity of knockout and complementation mutants were determined. It was found that the growth rate of *U. virens* was significantly reduced after the deletion of *UvALG3*, *UvALG15*, and *UvALG32*, indicating that the mannosyltransferase-encoding gene *UvALGs* plays a certain regulatory role in the vegetative growth of *U. virens*. There was no significant change in sporulation except for *UvALG3*, and the pathogenicity was significantly weakened, indicating that these three genes regulate the growth, sporulation, and pathogenicity of *U. virens* and are the pathogenic genes of the pathogen. Combining the protein domains encoded by these three genes, it was found that their protein domains were similar and their genetic relationships were similar. It is speculated that this unique domain is related to the pathogenicity of pathogens. Subsequently, the structural domains of these three proteins can be studied to further explore the relationship between the structure and function of mannosyltransferase.

In summary, this study preliminarily analyzed the function of the mannosyltransferase gene in *Ustilaginoidea virens* and provides genetic resources for the creation of new rice varieties resistant to *Ustilaginoidea virens*.

## 2. Results

### 2.1. Acquisition of Mannosyltransferase Gene of U. virens and Expression Quantity Analysis

Combined with the NCBI and CAZY databases, 38 mannosyltransferase-related genes were screened in the whole genome of *Ustilaginoidea virens*, and 38 mannosyltransferase genes were annotated (Table 1).

The qRT-PCR technique was utilized to determine the varying expression levels of *UvALG* genes at different stages of infection by *Ustilaginoidea virens* in rice (Figure 1A) and at various time points of *Ustilaginoidea virens*’s growth on medium(Figure 1B).

The gene expression levels of 38 *UvALGs* were analyzed, and three genes, namely, *UvALG3*, *UvALG15*, and *UvALG32*, exhibited significant upregulation during rice infection and *U. virens*’s growth on various days, warranting further investigation.

### 2.2. UvALGs’ Annotation and Bioinformatics Analysis

The number of amino acids, relative molecular weight, theoretical isoelectric point, instability coefficient, and hydrophobicity of the three proteins were analyzed online using the Ex PASy Proteomics Server website (http://web.expasy.org/protparam/, accessed on 15 September 2024). The results show that the amino acid length of the three proteins was about 400 aa, which was relatively stable, including two hydrophilic proteins and one water-transporting protein (Table 2 and Table 3). Other biological characteristics and phylogenetic trees are shown in Figure 2.

### 2.3. Isolation and Identification of UvALG Mutants and Complements

The upstream and downstream fragments of the knockout vector were amplified using the WT genomic DNA of wild-type *Ustilaginoidea virens* as a template, and the amplification results were observed by 1% agarose gel electrophoresis. After the knockout vector was successfully constructed, it was verified by double-enzyme digestion and sequencing, and the next test was carried out on the correct strain.

The knockout vector was introduced into *Agrobacterium* ALG-1, and the fast-growing single colony of *Agrobacterium* was selected from the transformed plate and cultured in liquid LB medium containing 50 µg/mL rifampicin and 50 µg/mL kanamycin at 28 °C for 3 days. The upstream and downstream fragment primers and hygromycin primers were used for PCR verification. The amplified bands of nine Agrobacterium AGL-1 were consistent with the expected bands, indicating that the knockout vectors PXEHA3, PXEHA15, and PXEHA32 had been introduced into *Agrobacterium* AGL-1 (Figure 3, Figure 4 and Figure 5).

#### 2.3.1. UvALG Mutation and Complementary Transformants Were Verified by RT-PCR

RNA was extracted from all knockout mutants and complemented mutants, and the RNA was reversed into cDNA using a reverse transcription kit. According to the mRNA sequence of the mannosyltransferase gene, primers were randomly selected from the 200–300 bp sequence. The primers β-tubulin F/β-tubulin R were designed according to the mRNA sequence of the β-tubulin gene of Aspergillus oryzae. The cDNA of the knockout and complemented mutants was used as a template, and the wild-type cDNA was used as a positive control. PCR amplification was performed using the cDNA of β-tubulin gene of Aspergillus oryzae as an internal reference (Figure 6).

#### 2.3.2. Pathogenicity Determination of Knockout and Complementation Mutant Strains

The results show that the disease was more serious after inoculation of wild-type strains, and the average number of rice balls per panicle was 20.33 ± 1.57. The average numbers of rice balls per panicle were 4.33 ± 0.57, 6.67 ± 1.15, and 5.53 ± 1.15 for the knockout transformants ΔUvALG3, ΔUvALG15, and ΔUvALG32, respectively. Compared with the wild-type and complementary transformants, the pathogenicity of the UvALG3 and UvALG32 gene deletion was significantly weakened, and the rice balls were reduced after UvALG15 gene knockout. The pathogenicity was weakened (Figure 7).

#### 2.3.3. Sensitivity of the Knockout and Complement Mutant Strains to Different Pressure Stresses Was Determined

Since the pathogenicity of *UvALG3*, *UvALG15*, and *UvALG32* genes was significantly weakened after knockout, the growth of these three genes under different stress conditions was measured. The growth of wild-type strain WT and *UvALG3*, *UvALG15*, and *UvALG32* knockout and complemented transformants on PSA medium was used as a control. After 14 days of culture on PSA medium containing 0.01% H_2_O_2_, 0.02% SDS, 120 µg/mL Congo Red (CR), 120 µg/mL Calcofluor White (CFW), 0.4 M sorbitol, and 0.25 M NaCl, the colony diameter was photographed and measured, and the inhibition rates of different gene knockout transformants under different stresses were calculated (Figure 8).

In conclusion, the deletion of *UvALG3*, *UvALG15*, and *UvALG32* genes affected the sensitivity of *U. virens* to osmotic stress, oxidative stress, and cell wall integrity to varying degrees, which may be related to the specific function of the genes.

## 3. Discussion

In Section 1, 38 genes encoding mannosyltransferases in *Ustilaginoidea virens* were screened by gene annotation. The expression levels of these genes were analyzed at different stages of *Ustilaginoidea virens* infection and different days of *Ustilaginoidea virens* growth on the medium. Three significantly upregulated genes encoding mannosyltransferases were screened and analyzed by bioinformatics. These three genes were *Uv8b_04477*, *Uv8b_01893*, and *Uv8b_04469*, respectively. The protein domains and physicochemical properties of these genes, the localization of coding genes on chromosomes, the prediction of protein functions, and the construction of phylogenetic trees were predicted and analyzed. It was found that the three proteins contained the following different domains: ALG3, GT15, and OCH1. The proteins were consistent with the corresponding homologous gene domains. The amino acid length of the three proteins was about 400 aa, which was relatively stable, including two hydrophilic proteins and one water-transporting protein. All chromosome sequences of *U. virens* were downloaded from NCBI. There were seven chromosomes in *U. virens*. *UvALG32* was located on the second chromosome, and *UvALG3* and *UvALG15* were located on the third chromosome. On the phylogenetic tree, the *UvALG3*, *UvALG15*, and *UvALG32* proteins were homologously aligned and phylogenetically analyzed with mannosyltransferases of fungi, such as Fusarium graminearum and Saccharomyces cerevisiae. The results show that *UvALG3* had the highest homology with *S. cerevisiae* ALG3 (96%), *UvALG15* had the highest homology with *S. cerevisiae* KTR1 (100%), and *UvALG32* had the highest homology with *F. graminearum* OCH1 (100%).

In order to further study the function of the mannosyltransferase-encoding genes *UvALG3*, *UvALG15*, and *UvALG32*, based on the vector PXEH, the gene knockout vectors PXEHA3, PXEHA15, and PXEHA32 of *UvALG3*, *UvALG*15, and *UvALG32* were constructed by *Agrobacterium-mediated* genetic transformation method. The knockout transformants were obtained by *Agrobacterium-mediated* transformation and verified by PCR experiments. Finally, the knockout transformants ΔUvALG3, ΔUvALG15, and ΔUvALG32 were selected for subsequent experiments. Based on the vector PXNPTII, the *UvALG3*, *UvALG15*, and *UvALG32* complementation vectors CPXEHA3, CPXEHA15, and CPXEHA32 were constructed. Using the same techniques and methods as the *UvALG3*, *UvALG15*, and *UvALG32* knockouts, the *UvALG3*, *UvALG15*, and *UvALG32* complementary transformants CΔUvALG3, CΔUvALG15, and CΔUvALG32 were finally obtained. Gene function verification experiments showed that the deletion of *UvALGs* changed their vegetative growth compared with the wild-type rice false smut strain WT, and the growth rate of knockout mutants on PSA medium significantly slowed down. This indicates that the mannosyltransferase-encoding genes’ UvALGs play a certain regulatory role in the vegetative growth of *Ustilaginoidea virens*.

The N-glycosylation pathway of *S. cerevisiae* is widely conserved, but the biological functions of N-glycosylation are not well characterized in filamentous fungi, particularly in fungal plant pathogens [27,34]. In this study, the ALG3 gene was identified by insertional mutagenesis as an important virulence factor in *U. virens*. ALG3 encodes an ER protein orthologous to yeast Alg3. When expressed in *S. cerevisiae*, the *U. virens* ALG3 functionally complemented the temperature sensitivity of an alg3 stt3 double mutant. Furthermore, we showed that oligosaccharides linked to Slp1 or CPY in the ALG3 deletion mutant could not be digested by the enzyme Endo H [10,35,36]; thus, it can be postulated that they are Man5-oligosaccharides but not Man9-oligosaccharides in the wild type [37,38,39,40]. Therefore, in the same way as its yeast ortholog, ALG3 encodes an a-1,3-mannosyltransferase that is involved in the biosynthesis of N-glycan for the N-glycosylation of proteins in *U. virens*.

The ALG gene was knocked out and complemented, and the growth rate, sporulation, and pathogenicity of the knockout and complementary transformants were determined. It was found that after the deletion of *UvALG3*, *UvALG15,s* and *UvALG32*, the growth rate of *U. virens* was significantly reduced, the sporulation significantly increased, except for *UvALG32*, and the pathogenicity was significantly weakened or lost, indicating that these three genes regulate the growth, sporulation, and pathogenicity of *U. virens* as pathogenic genes of the pathogen. Combined with the protein domains encoded by these three genes, it was found that their protein domains were similar, and their genetic relationships were similar. It is speculated that this unique domain is related to the pathogenicity of pathogens. Subsequent studies can be conducted on the domains of these three proteins to further explore the relationship between the structure and function of mannosyltransferase.

## 4. Materials and Methods

### 4.1. Strains and Culture Conditions

The two representative *U. virens* strains used in this study were isolated from infected Oryza sativa in Shenyang and Dandong in the Liaoning Province of China, respectively. which was used to generate all mutant strains. The isolates were placed in potato sucrose agar (PSA) medium at 25 ± 1 °C. After 7 days of cultivation, the mycelium on the outer edge of the colony was collected using an inoculation needle and placed in a new PSA medium.

### 4.2. Analysis of UvALGs’ Gene Expression

Rice panicle samples were collected at 1 d, 3 d, 6 d, 9 d, 14 d, and 21 d after inoculation with wild-type *U. virens* WT, and samples of U. virens cultured on PSA medium for 7 d, 14 d, and 21 d were collected. After quick freezing with liquid nitrogen, RNA was extracted from the samples, and the extracted RNA was reversed into cDNA (Primers in Table 4).

(1) Extraction of total RNA

The liquid-nitrogen quick-frozen mycelium preserved in a −80 °C ultra-low-temperature refrigerator was removed, and 5 g of each sample was weighed and placed in a mortar treated with 0.1% diethyl pyrocarbonate (DEPC) solution in advance, and ground into powder with liquid nitrogen. The entire test procedure needs to be carried out in a clean environment, and masks and disposable gloves must be worn; 1 mL RNAiso Plus was added to the mortar and left at room temperature until the powder was lysed by the RNA extract becoming clear and transparent. The supernatant of the homogenized pyrolysis liquid in the mortar was transferred to a 1.5 mL centrifuge tube of RNase-free, and a one-fifth volume of chloroform was added. The mixture was shaken and mixed until the solution was emulsified to milky white and left on ice for 5 min. It was centrifuged at 12,000 r/min and 4 °C for 15 min. The supernatant was transferred to a new 1.5 mL RNase-free centrifuge tube, and an equal volume of anhydrous ethanol was added. It was mixed upside down, and allowed to stand at −20 °C for 1 h. After centrifugation at 12,000 r/min for 10 min at 4 °C, the supernatant was discarded. A total of 1 mL 75% ethanol was added, and the precipitate was washed upside down; it was centrifuged at 12,000 r/min at 4 °C for 10 min, and then the supernatant was discarded and the liquid in the centrifuge tube was drawn out with a special RNA gun head. The centrifuge tube’s cover was opened, and the precipitate was dried in a clean environment, with an appropriate amount of RNase-free dd H_2_O added to dissolve the precipitate. It was stored at−80 °C. The extracted RNA could be detected by 1% agarose gel electrophoresis and an ultramicro spectrophotometer for concentration and quality.

(2) Synthesis of cDNA

The cDNA was synthesized using the Easy Script One-Step g DNA Removal and c DNA Synthesis Super Mix. The specific steps were: 1–5 µg Total RNA, 1 µL RandomPrimer (N9) (0.1 µg/µL), 10 µL 2 × ES Reaction Mix, 1 µL Easy Script ^®^ RT/RI Enzyme Mix, 1 µL g DNA Remover, and RNase-free dd H_2_O to 20 µL were added to the RNase-free PCR tube in turn. The solution was gently mixed, incubated at 25 °C for 10 min, and then incubated at 42 °C for 30 min. The Easy Script ^®^ RT/RI and g DNA Remover in the tube were inactivated by heating at 85 °C for 5 s on the PCR instrument(Hangzhou Bori Technology Co., Ltd. (Hangzhou, China) TC-96/G/H(b)C) and then stored in the refrigerator at −20 °C.

(3) Quantitative Real-time polymerase chain reaction

Real-time fluorescence quantitative PCR. The β-tubulin gene of A. oryzae was used as an internal reference. The cDNA of rice samples at 1 d, 3 d, 6 d, 9 d, 13 d, and 21 d after inoculation was used as a template for the PCR amplification using a real-time fluorescence quantitative PCR kit. The steps were as follows: About 1 µg template, 0.4 µL forward primer (10 µM), 0.4 µL reverse primer (10 µM), and 10 µL 2 × Perfect Start ^®^ Green q PCR Super Mix were added to the PCR 96-well plate according to the reaction system, and nuclease-free water was added to a total volume of 20 µL. After configuration, it was gently aspirated and mixed, and the 96-well plate and centrifuge were sealed to ensure the liquid gathered at the bottom of the tube. The configuration was placed on ice throughout the process. The amplification reaction was performed on a Bio-Rad CFX96 instrument. Reaction conditions: 94 °C, 30 s; 94 °C, 5 s; and 60 °C, 30 s; repeated for 35–50 cycles. After amplification, the Ct value of each reaction was recorded, and the relative expression of the gene was calculated using the 2^−ΔΔCt^ method.

### 4.3. Gene Disruption and Complementation of the UvALGs

The *UvALGs* sequences were used as templates, and the upstream and downstream homologous arms of the knockout vectors were designed using Primer 5 and DNAMAN 6.0.3.48 to amplify specific primers. Combined with the pXEH vector sequence, the enzyme digestion site and homologous arm were added to the 5’ end of the specific primer. The upstream restriction sites of the *UvALGs* knockout vectors were *EcoR* I and *Kpn* I, and the downstream restriction sites were *Xba* I and *Hind* III. The vector construction strategy was to first connect the upstream fragment and then connect the downstream fragment. Flanking sequences of about 1000 bp at both ends of the target gene were amplified by PCR, and the two flanking sequences were connected to the pXEH vector.

#### 4.3.1. Acquisition of Flanking Sequences F1 and F2

The target gene sequence of the family gene and the sequence of the two ends of the target gene were found in the genome of Aspergillus oryzae. The forward and reverse primers UF/UR and DF/DR of the gene flanking sequences F1 and F2 were designed by Primer 5.0 software. At the same time, the enzyme digestion site used in the connection was selected, and the connector was added at both ends of the primer. When selecting the enzyme cutting site, the one contained in the first fragment of the connection should be avoided. The primer sequence and amplification system were as follows: DNA 1 µL, 2 × Phanta Max Buffer, 25.0 µL; dNPT Mix (10 mM), 1.0 µL; upstream primer (10 mM), 2.0 µL; downstream primer (10 mM), 2.0 µL; Phanta Max Super-Fidelity DNA Polymerase, 1.0 µL; and ddH_2_O to a volume of 50 µL. Reaction procedure: 95 °C, 3 min; 95 °C, 15 s; tm 60 °C, 15 s; 72 °C, 1 min, a total of 35 cycles; 72 °C, 5 min; 12 °C, ∞. 

#### 4.3.2. Linearized Carrier

The purified empty vector pXEH plasmid was double digested according to the following system and reaction procedure. The linearized vector was separated by 1.0% agarose gel electrophoresis. The enzyme digestion system was as follows: fast-cutting enzyme I, 1 µL: fast-cutting enzyme II, 1 µL; rCut Buffer, 5 µL; plasmid DNA, 5 µL or 10 µL; and ddH_2_O to a volume of 50 µL. Reaction procedure: 37 °C for 30–45 min.

#### 4.3.3. Gel Recovery and Purification

The correct bright and single band was cut into the sterilized 2 mL EP tube, and the corresponding volume of the binding buffer was added. The binding buffer, at 65 °C, was hot melted for 8 min, and every 2 min the EP tube was inverted and mixed to accelerate the gel’s melting. After fully dissolving, it was poured into the Hindband adsorption column, centrifuged at 13,000 rpm for 1 min at room temperature, and the waste liquid was discarded (during this step, the waste liquid can be appropriately poured back into the adsorption column and centrifuge). Adding 700 µL of spw wash buffer, it was centrifuged at the maximum speed for 1 min, and the waste liquid was discarded. This was repeated once. The empty column was centrifuged at a maximum speed of 3 min. The adsorption column was transferred to a clean 1.5 mL EP tube, aired under a working table, and dried with alcohol for 10 min. At this time, the EB was preheated in a 65 °C metal bath. Adding 30–50 µL of preheated EB, it was left to stand for 2 min and then centrifuged at the maximum speed for 1 min. The concentration was measured and stored at −20 °C.

#### 4.3.4. Carrier Recombination Reaction System and Escherichia Coli Transformation

Linearized carrier, 1.5 µL; insertion element, 3.5 µL; 5×CE II Buffer, 2 µL; Exnase II, 0.5 µL; and ddH_2_O, 2.5 µL. The reaction procedure: 37 °C for 40–60 min. Ten minutes before the completion of the carrier recombination reaction, the DH5α competent state was removed from the −80 °C ultra-low-temperature refrigerator and placed on ice for static melting. Adding 5 µL of the carrier connection product to 50 µL of DH5α competent liquid, it was left to stand on ice for 30 min; at 42 °C for heat shock for 45 s; and then an ice water bath for 2 min. In an ultraclean working state, 700 µL LB liquid medium was added, and at 37 °C and 180 rpm, it was cultured for 1–2 h (a solid LB medium containing the corresponding antibiotics was prepared) and then centrifuged at 8000 rpm for 1 min. Removing 700 µL of supernatant, it was aspirated and mixed well, taking 100 µL of bacterial liquid and evenly coating it on the pre-prepared LB solid medium containing the corresponding antibiotics and then cultured at 37 °C for 12–20 h. Single colonies were collected and placed in PCR tubes containing 20 µL LB and mixed well for use as PCR template DNA. PCR was performed, and the system and reaction procedure were as follows: DNA, 1 µL; forward primer, 0.5 µL; reverse primer, 0.5 µL; mix, 10 µL; and ddH_2_O 8 µL. Reaction procedure: 95 °C, 10 min; 95 °C, 15 s; tm 60 °C, 15 s; and 72 °C, 65 s for a total of 35 cycles; 72 °C, 5 min; 16 °C, ∞. Taking 5 µL of the bacterial solution corresponding to the bright single band, 5 mL of liquid LB medium containing 5 µL of antibiotics was added, and the bacteria was shaken at 37 °C and 180 rpm in an overnight culture.

#### 4.3.5. Plasmid Extraction

The above bacterial solution was added to a 2 mL EP tube and centrifuged at 8000 rpm for 1 min, and the waste liquid was discarded until the cell collection was completed. Adding 250 µL SI solution, it was vortexed until the bacteria melted. Adding 250 µL SII solution, it was mixed well and left to stand for 1 min. Adding 350 µL SIII solution, it was mixed well at 13,000 rpm for 10 min. Adding the supernatant to the adsorption column, it was centrifuged at 13,000 rpm for 1 min, and the waste liquid was discarded (this step can be repeated as appropriate). Adding 700 µL DNA wash buffer, it was centrifuged at 13,000 rpm for 1 min. Discarding the waste liquid, the empty column was centrifuged for 3 min. Discarding the collection tube, the adsorption column was transferred to a clean 1.5 mL EP tube and aspirated and dried by alcohol in an ultra-clean bench for 10 min. Preheating the EB (65 °C), 30 µL was added, left to stand for 2 min, and centrifuged at 13,000 rpm for 1 min. After measuring the concentration, it was stored at −20 °C for preservation.

#### 4.3.6. Verification of Vector Restriction Enzyme Digestion of the Upstream Fragment

The vector connected to the upstream fragment was verified by the following enzyme digestion system and reaction procedure: 1% agarose gel electrophoresis. The enzyme digestion system was as follows: fast-cutting enzyme I, 1 µL; fast-cutting enzyme II, 1 µL; rCut Buffer, 2 µL; plasmid DNA, 3 µL; and ddH_2_O, 8 µL. The reaction procedure was as follows: 37 °C for 30–45 min.

#### 4.3.7. Knockout of the Connection of the Downstream Fragment of the Vector

The vector correctly connected to the upstream fragment was linearized with *Xba* I and *Hind* III double digestion, and then the linearized vector and the downstream fragment were recovered by gel cutting, the vector recombination reaction, Escherichia coli transformation, and plasmid extraction test scheme were the same as above.

#### 4.3.8. Validation of Knockout Vectors

The constructed knockout vector was verified by double-enzyme digestion, and the enzyme digestion site and fragment size are presented in Table 5.

#### 4.3.9. Agrobacterium Competent Preparation and Agrobacterium Transformation

*Agrobacterium* AGL-1 was taken from the −80 °C ultra-low-temperature refrigerator and cultured on LB solid medium containing 50 µg/mL rifampicin for 3 days at 28 °C. A single colony of Agrobacterium AGL-1 was selected and cultured in LB liquid medium containing 50 µg/mL rifampicin, and at 28 °C and 180 rpm it underwent shaking culture for 40 h to an OD660 = 0.4–0.6. It was left on ice for 30 min, and at 4 °C and 8000 rpm it was centrifuged for 5 min, removing supernatant as much as possible. The cells were resuspended with an equal volume of 20 mM CaCl_2_ at 4 °C and 8000 rpm for 5 min. The supernatant was discarded, and the bacteria were retained and mixed with a 1/10 volume of 20 mM CaCl_2_ (containing 20% glycerol). A 100 µL suspension was taken and divided into 1.5 mL EP tubes, frozen in liquid nitrogen, and stored at −80 °C. Melting Agrobacterium-competent AGL-1 on ice 10 min in advance, 5 µL recombinant plasmid was added to the sub-packed 50 µL of the Agrobacterium-competent liquid surface, and left to stand on ice for 30 min. The EP tube was frozen in liquid nitrogen for 2 min, immediately placed in a 37 °C water bath for 3 min, and then quickly placed on ice for 3 min. Then, 700 µL liquid LB medium was added, and at 28 °C and 180 rpm it underwent shaking culture for 3–4 h. At 8000 rpm, it was centrifuged for 1 min/5000 rpm and then 3 min, discarding 600 µL of supernatant. The bacteria was aspirated and mixed evenly, coating the suspension on an LB solid medium (50 µg/mL RIF and 50 µg/mL Kan) with two antibiotics, and cultured for 3 days at 28 °C.

#### 4.3.10. Agrobacterium Verification

A single colony was selected and cultured in 5 mL LB liquid medium containing 50 µg/mL RIF and 50 µg/mL Kan at 28 °C and 180 rpm for 3 days.

The following primers were selected to verify the upstream and downstream flanking fragments and hygromycin gene (Table 6).

The PCR system and reaction procedure were the same as above.

#### 4.3.11. Screening of the Minimum Tolerance of Ustilaginoidea virens to Hygromycin

In order to explore the resistance of different hygromycin concentrations to Ustilaginoidea virens, five PSA mediums with different hygromycin concentrations of 0 µg/mL, 200 µg/mL, 400 µg/mL, 600 µg/mL, and 800 µg/mL were set up, and three replicates were set for each concentration. After 14 days of culture, the inhibition of different hygromycin concentrations on the mycelial growth of *U. virens* was observed, and the minimum tolerance concentration of *Ustilaginoidea virens* to hygromycin was screened.

#### 4.3.12. Preparation of Conidia of Ustilaginoidea virens

After 14 days of inverted culture on PSA medium at 28 °C, the conidia of *Ustilaginoidea virens* were scraped off with IM liquid medium and a sterile scalpel on the ultra-clean bench. The spore suspension was filtered with sterile three-layer filter paper. The spore concentration was adjusted to 10^6^ spores/mL (about 20 spores under a 40 times microscope), and the germinated conidia were cultured at 28 °C for 6 h.

#### 4.3.13. Agrobacterium Mediated Genetic Transformation of Ustilaginoidea virens

The correct *Agrobacterium* was stored at −80 °C with 40% glycerol. The activated Agrobacterium (50 µg/mL RIF and 50 µg/mL Kan) was centrifuged at 6000 rpm for 5 min, the supernatant was discarded, and the precipitate was resuspended with IM-induced medium, centrifuged at 6000 rpm for 5 min, repeated 3 times, to remove as much as possible antibiotics. The bacterial solution was adjusted to OD660 = 0.2 with IM liquid medium, and 5 mL was transferred to a 50 mL centrifuge tube to a final concentration of 200 µM AS (AS was taken out half an hour in advance and allowed to stand on ice). It was shake cultured at 28 °C and 200 rpm to OD660 = 0.5. Then, 1 mL Agrobacterium was transferred to a 2 mL centrifuge tube, adding 1 mL of spore suspension and co-cultivating for 2 h. The sterilized cellophane was spread on the IM solid medium, and 200 µL of the mixture was placed in the center of the cellophane, coated with a coater to a slight resistance, and cultured at 28 °C in inverted darkness for 48 h. Then, the cellophane was transferred to PSA containing 300 µg/mL cephalosporin and 200 µg/mL Hyg, and cultured at 28 °C inverted for 14 d.

#### 4.3.14. RT-PCR Verification of Knockout Mutants and Complemented Mutants

In order to verify whether the true complementary transformants were obtained and whether the target gene could be expressed normally in the complementary transformants, RT-PCR experiments were carried out. The mRNA sequence of the target gene of *Ustilaginoidea virens* was analyzed. Primers P9/P10 were designed on the mRNA fragment with Primer 5.0 software. The PCR product was about 300 bp in size. The cDNA of three knockout transformants and three complementary transformants of the wild-type strain were used as templates for the PCR amplification. The wild-type strain could amplify the band, the knockout transformant could not amplify the band, and the complement transformant could amplify the band with the same size as the wild type. At the same time, the primers β-tublin F/β-tublin R were designed on the mRNA fragment of the β-tublin gene by Primer 5.0 software with the β-tublin gene as the internal reference. The cDNAs of the wild-type strain, three knockout transformants, and three complement transformants were used as templates for the PCR amplification.

## Figures and Tables

**Figure 1 ijms-26-02979-f001:**
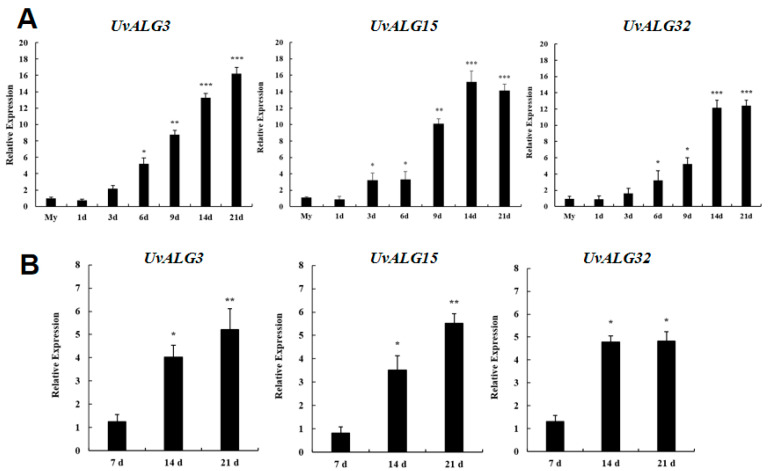
Expression analysis of mannosyltransferase: (**A**) expression analysis of *UvALGs* in different stages of rice infected with *U. virens*; (**B**) expression analysis of *UvALGs* in *U. virens* grown on medium for different days. (* = 0.01 < *p* < 0.05, ** = *p* < 0.01, *** = *p* < 0.001).

**Figure 2 ijms-26-02979-f002:**
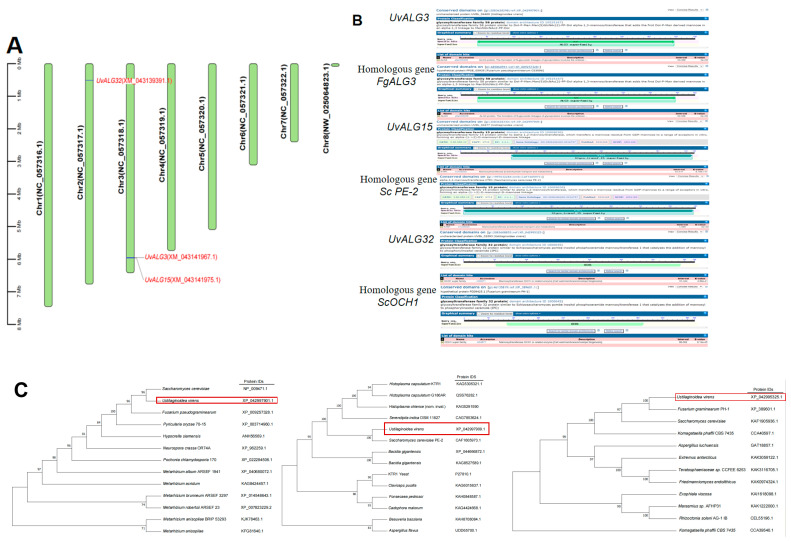
ALGs bioinformatics analysis of *U. virens*: (**A**) *UvALG32* gene was located on the second chromosome, and *UvALG3* and *UvALG15* were located on the third chromosome, as shown in Figure 2; (**B**) three mannosyltransferases have conserved domains—ALG3, GT15, and OCH1—that are consistent with homologous genes; (**C**) in order to further clarify the function of the three genes, the *UvALG3*, *UvALG15*, and *UvALG32* proteins were compared with the mannose transferases of *Fusarium graminearum* and *Saccharomyces cerevisiae*. The results show that *UvALG3* had the highest homology with *S. cerevisiae ALG3* (96%), *UvALG15* had the highest homology with *S. cerevisiae* KTR1 (100%), and *UvALG32* had the highest homology with *F. graminearum* OCH1 (100%).

**Figure 3 ijms-26-02979-f003:**
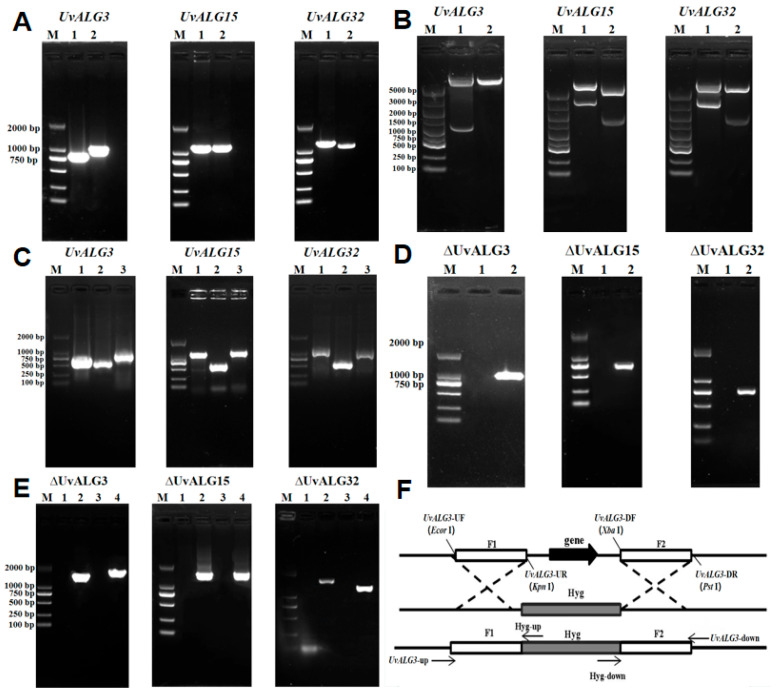
Construction of ALG knockout vectors of *U. virens*. (**A**) Acquisition of upstream and downstream fragments of knockout vectors: (M) 2K DNA Maker, (1) upstream fragment, (2) downstream fragment. (**B**) Knockout vector enzyme digestion verification: (M) 5K DNA Maker, (1) knockout vector; (2) PXEH vector. (**C**) PCR verification of *Agrobacterium tumefaciens* transformation: (M) 2K DNA Maker, (1) upstream segment, (2) HYG internal gene, (3) downstream fragment. (**D**) PCR verification of ∆UvALG knockout transformants: (M) 2K DNA Maker, (1) ∆UvALGs, (2) WT (wild type). (**E**) Upstream and downstream PCR verification of ΔUvALG knockout transformants: (1,3) WT (wild type) and (2,4) ΔUvALGs. (**F**) The *UvALGs*’ sequences were used as templates, and the upstream and downstream homologous arms of the knockout vectors were designed using Primer 5 and DNAMAN 6.0.3.48 to amplify specific primers. Combined with the pXEH vector sequence, the enzyme digestion site and homologous arm were added to the 5’ end of the specific primer. The upstream restriction sites of the *UvALG* knockout vectors were *EcoR* I and *Kpn* I, and the downstream restriction sites were *Xba* I and *Hind* III. The vector construction strategy was to first connect the upstream fragment and then connect the downstream fragment. The specific strategy is as shown in the figure. The flanking sequence of about 1000 bp at both ends of the target gene was amplified by PCR, and the two flanking sequences were connected to the pXEH vector.

**Figure 4 ijms-26-02979-f004:**
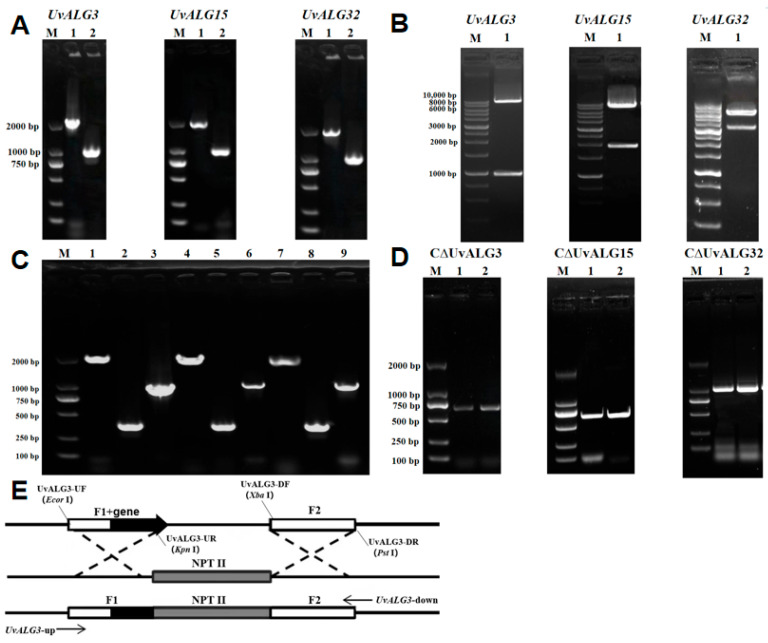
Construction of ALG gene complementation vectors of *U. virens*. (**A**) Acquisition of upstream and downstream fragments of the complement vector: (M) 2K DNA Maker, (1) upstream fragment, and (2) downstream fragment. (**B**) Complement vector enzyme digestion verification: (M) 10K DNA Maker and (1) complement vector. (**C**) PCR verification of Agrobacterium tumefaciens transformation: (M) 2K DNA Maker, (1) upstream segment, (2) NPT-II internal gene, and (3) downstream fragment. (**D**) PCR verification of C∆UvALG knockout transformants: (M) 2K DNA Maker, (1) C ∆UvALGs, and (2) WT (wild type). (**E**) Construction strategy of the complementary carrier.

**Figure 5 ijms-26-02979-f005:**
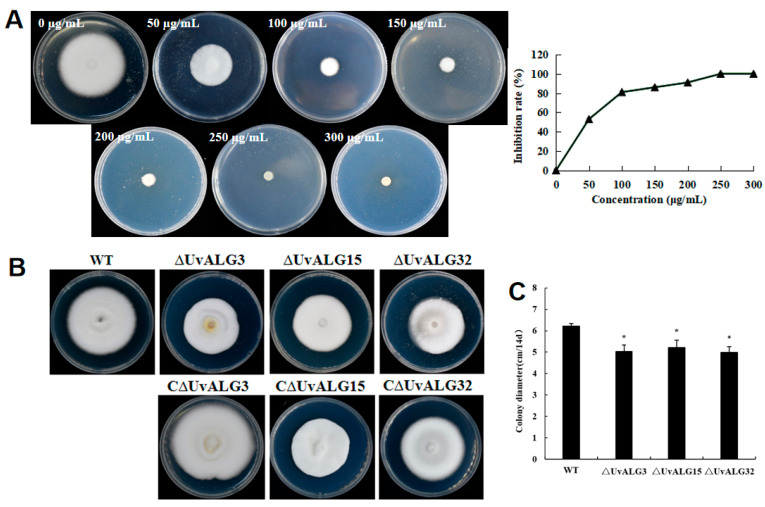
Hygromycin resistance screening and mutant acquisition in *U. virens*. (**A**) Screening of the minimum tolerance concentration of hygromycin to *U. virens*. Hygromycin PSA culture with different concentrations (0 µg/mL, 50 µg/mL, 100 µg/mL, 150 µg/mL, 200 µg/mL, 250 µg/mL, and 300 µg/mL) was inoculated on Aspergillus oryzae cake with a diameter of 5 mm. Three replicates were set for each concentration, and the growth of *Ustilaginoidea virens* at different concentrations was observed after 14 days in culture at a constant temperature of 28 °C. The results show that there were different degrees of growth at 0 µg/mL and 200 µg/mL. Finally, the tolerance of *Ustilaginoidea virens* to hygromycin was 250 µg/mL. (**B**) ΔUvALG and CΔUvALG colony morphologies. The co-transformed strains were grown on glass paper on PSA medium (300 µg/mL cephalosporin and 250 µg/mL hygromycin B) for 7 days, and single colonies were picked and subcultured on new PSA medium. After single spore isolation, the purified strains were selected from the new PSA medium to obtain three mutants of ∆UvALG3, ∆UvALG15, and ∆UvALG32 genes. Complementary vector screening replaced the hygromycin resistance fragment with G418. (**C**) Gene knockout and complement mutant colony diameter. The WT strain and all gene knockout and complemented mutants were cultured on PSA medium for 14 days and the colony diameter was measured. After 7 days of shaking culture in PS medium, the sporulation was calculated using a blood cell counting plate. The colony’s morphology is shown in the image. The results show that after 14 days of culture, the average colony diameter of the WT strain was 6.13 ± 0.05 cm, and the average colony diameters of knockout mutants ΔUvALG3, ΔUvALG15, and ΔUvALG32 were 4.80 ± 0.31 cm, 5.21 ± 0.29 cm, and 4.79 ± 0.21 cm, respectively. Compared with the WT strain, the growth rate was significantly reduced. (* = *p* < 0.05).

**Figure 6 ijms-26-02979-f006:**
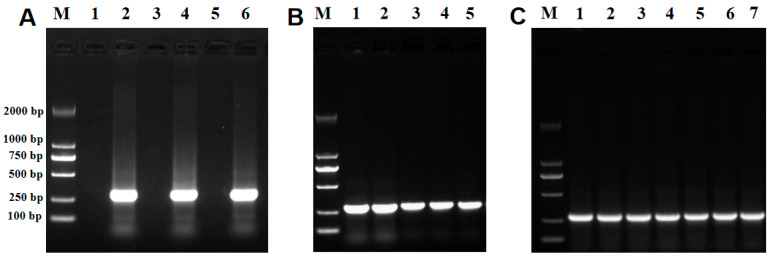
RT-PCR verification of *UvALG* mutants and complements. (**A**) Knockout target gene verification: (1.3.5) ∆UvALGs’s target gene and (2.4.6) WT (wild type). (**B**) Complementary target gene verification: (1.2.3) C∆UvALG 3, C∆UvALG 15, and C∆UvALG 32 and (4.5) WT (wild type). (**C**) Verification of the β-tubulin reference gene in *Ustilaginoidea virens*: (1.2.3) ∆UvALG3, ∆UvALG15, and ∆UvALG32; (4.5.6) CΔUvALG3, CΔUvALG15, CΔUvALG32, and (7) WT (wild type); and (M) 2K Maker.

**Figure 7 ijms-26-02979-f007:**
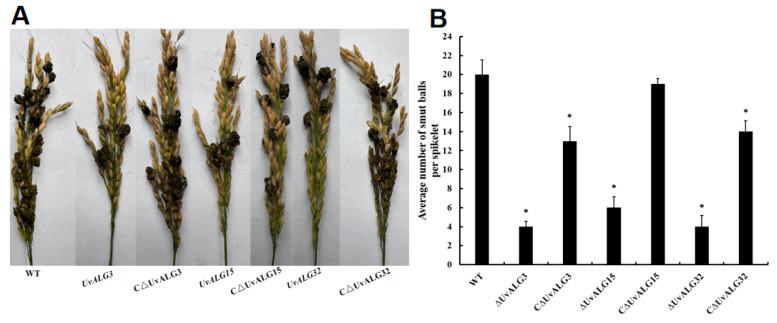
*UvALG* knockout and complementation transformants were inoculated on rice. (**A**) Incidence of *UvALG* knockout and complementation transformants. The wild-type strain, WT, mannosyltransferase gene knockout and complementation transformants of Aspergillus oryzae were cultured in PSB medium for 7 days, ground into homogenate, adjusted for concentration, and inoculated with Liaoyou 65. After 21 days, the incidence was investigated and the number of rice balls was counted. The incidence is shown in the figure. (**B**) Pathogenicity of *UvALG* knockout and complementary transformants. * *p* < 0.05.

**Figure 8 ijms-26-02979-f008:**
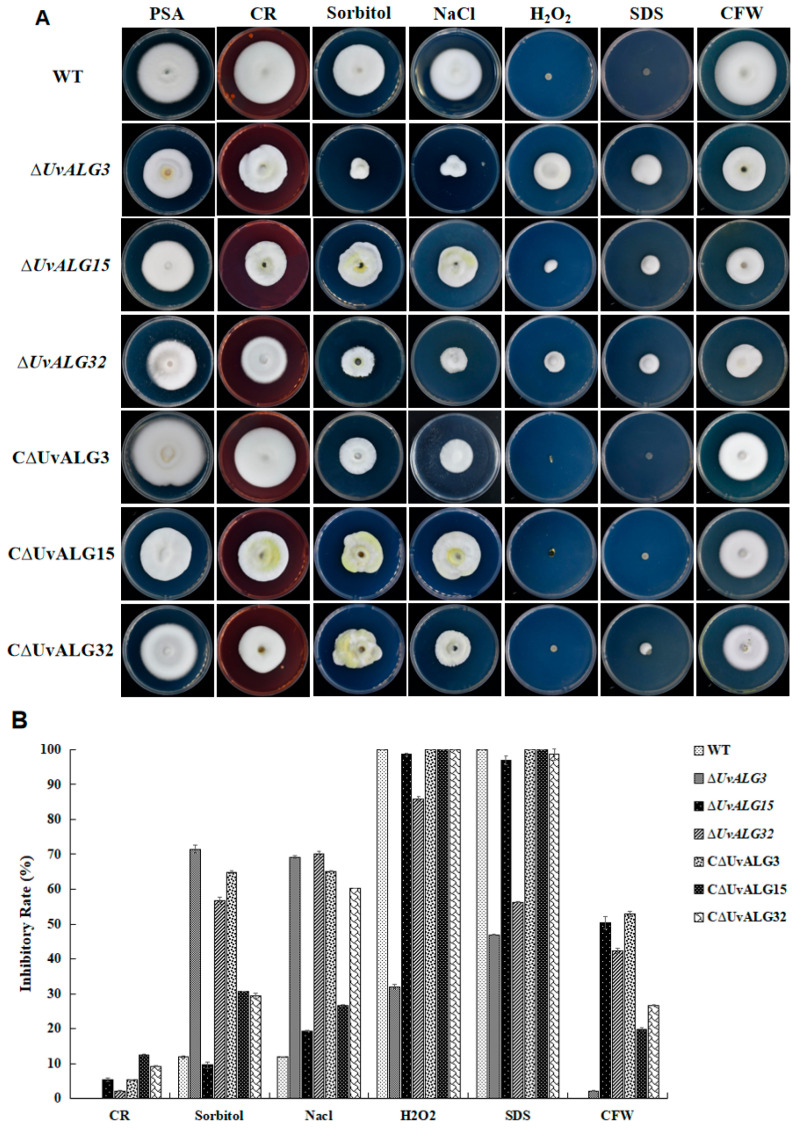
Effects of different chemical stresses on strains. (**A**) Growth of *UvALG* knockout and complementation transformants under different stress conditions. (**B**) For the *UvALG3* and *UvALG32* genes, the inhibition rate of 120 µg/mL CR PSA medium on the mycelial growth of knockout transformants was almost the same as that of wild-type strains WT and CΔUvALG3 strains. The inhibition rate of 120 µg/mL CFW PSA medium on the mycelial growth of the knockout transformants was significantly lower than that of the wild-type strains WT and CΔUvALG3 strains. Under the pressure stress of 0.4 M sorbitol and 0.25 M NaCl, the inhibition rate of mycelial growth of the knockout transformants was significantly lower than that of the wild-type strains WT and CΔUvALG3 strains. On PSA medium containing 0.015% H_2_O_2_ and 0.02% SDS, the mycelial growth inhibition rate of knockout transformants was higher than that of WT and CΔUvALG3. The results showed that the deletion of *UvALG3* and *UvALG32* genes significantly affected the sensitivity of *U. virens* to different osmotic pressures and cell wall integrity. Under the stress of 120 µg/mL CFW, it became more insensitive, and under the stress of 0.4 M sorbitol and 0.25 M Na Cl, it became more sensitive. The difference is that *UvALG15* under the stress of 0.4 M sorbitol and 0.25 M NaCl, the mycelial growth inhibition rate of knockout transformants and the growth inhibition rate of wild-type strains WT and CΔUvALG15 strains were not significantly different.

**Table 1 ijms-26-02979-t001:** Screening of mannosyltransferase genes in *Ustilaginoidea virens*.

Gene Encodes	Annotation	Domain
*Uv8b_07818*	Alpha 1,2 mannosyltransferase	Glyco_transf_15
*Uv8b_06540*	Dolichyl-phosphate-mannose-protein	PMT_2
*Uv8b_05581*	alpha-1,3/1,6-Mannosyltransferase ALG2 and similar proteins	GT4_ALG2-like
*Uv8b_04477*	alpha-1,2-Mannosyltransferase	Glyco_transf_15
*Uv8b_03749*	Dolichyl-phosphate-mannose-protein mannosyltransferase	PMT_2
*Uv8b_03634*	Dolichyl-phosphate-mannose-protein	PMT_2
*Uv8b_03269*	Mannosyltransferase OCH1 or related enzyme	OCH1
*Uv8b_03119*	alpha 1,2-Mannosyltransferase	Glyco_transf_15
*Uv8b_02986*	alpha-1,3-Mannosyltransferase CMT1	CAP59_mtransfer
*Uv8b_02466*	Dolichyl-phosphate-mannose-protein	PMT_2
*Uv8b_01893*	Mannosyltransferase OCH1 or related enzyme	OCH1
*Uv8b_01686*	Dolichyl-phosphate-mannose-protein	PMT_2
*Uv8b_01474*	Mannosyltransferase OCH1 or related enzyme	OCH1
*Uv8b_01342*	alpha-1,2-Mannosyltransferase ALG11	GT4_ALG11-like
*Uv8b_00803*	Dolichol-phosphate-mannosyltransferase subunit 3 (DPM3)	DPM3
*Uv8b_00648*	Dolichyl-phosphate-mannose-protein	PMT_2
*Uv8b_00068*	Dolichyl-phosphate-beta-D-mannosyltransferase	PLN02726
*Uv8b_07818*	alpha-1,2-Mannosyltransferase	Glyco_transf_15
*Uv8b_06540*	Dolichyl-phosphate-mannose-protein	PMT_2
*Uv8b_05581*	alpha-1,3/1,6-Mannosyltransferase ALG2	GT4_ALG2-like
*Uv8b_04467*	alpha-1,2-mannosyltransferase	Glyco_transf_15
*Uv8b_03749*	Dolichyl-phosphate-mannose-protein	PMT_2
*Uv8b_03269*	Mannosyltransferase OCH1 or related enzyme	OCH1
*Uv8b_03119*	alpha-1,2-Mannosyltransferase	Glyco_transf_15
*Uv8b_04469*	alpha-1,3-Mannosyltransferase	ALG3 super family
*Uv8b_02986*	Cryptococcal mannosyltransferase 1	CAP59_mtransfer
*UvI_02025670*	Dolichyl-phosphate-mannose-protein	PMT_2
*UvI_02035730*	Mannosyltransferase OCH1 or related enzyme	PRK08315 OCH1
*UvI_02008980*	alpha-1,2-Mannosyltransferase	Glyco_transf_15
*UvI_02020590*	Cryptococcal mannosyltransferase 1	CAP59_mtransfer
*UvI_02055610*	alpha-1,2-Mannosyltransferase	Glyco_transf_15
*UvI_02045040*	lpha-1,3/1,6-Mmannosyltransferase ALG2	GT4_ALG2-like
*UvI_02019520*	Mannosyltransferase OCH1	OCH1
*UvI_02012570*	Dolichol-phosphate mannosyltransferase subunit 3 (DPM3)	DPM3
*UvI_02003470*	alpha-1,2-Mannosyltransferase	Glyco_transf_15
*UvI_02058330*	alpha-1,2-Mannosyltransferase ALG11	GT4_ALG11-like
*UvI_02029900*	Mannosyltransferase OCH1 or related enzyme	OCH1
*UvI_02060380*	Dolichyl-phosphate beta-D-mannosyltransferase	PLN02726

**Table 2 ijms-26-02979-t002:** Open reading frame analysis of the *UvALGs*.

Gene	Orthologous Genes	Base Number	Intron	Number of AA
*UvALG3*	*FgALG3*	1416	2	471
*UvALG15*	*Sc PE-2*	1248	2	415
*UvALG32*	*ScOCH1*	1101	1	366

**Table 3 ijms-26-02979-t003:** Physicochemical properties of proteins encoded by the *UvALGs* of *Ustilaginoidea virens*.

Locus ID	Gene Name	Amino Acid Length/aa	Molecular Weight/kDa	Theoretical PI	Instability Index	Grand Average of Hydropathicity
>XP_042997901.1	*UvALG3*	435	48,986.95	9.81	33.75	0.426
>XP_042997909.1	*UvALG15*	415	48,270.18	5.92	40.7	−0.555
>XP_042995325.1	*UvALG32*	366	41,467.94	5.76	40.14	−3.634

**Table 4 ijms-26-02979-t004:** UvALGs’ qRT-PCR primers.

Primer Name	Primer Sequence 5′-3′
β- tubulin F	TCGTCGATTTGGAGCCTGGT
β- tubulin R	ATCTTGGAGATGAGCAAGGT
UvALG3-F	TCTAGACCTACTGCAGCATA
UvALG3-R	ACCAGAAGCAACGATTCTAT
UvALG15-F	ATTTCGCTGCTTCACGGCCT
UvALG15-R	GCTGAAGGAGCCATTGTAAA
UvALG32-F	TGATAATGACCCGATGGATT
UvALG32-R	ATTGTGATGATGACTCCCTT

**Table 5 ijms-26-02979-t005:** Knockout vector enzyme digestion verification.

Vector	Restriction Enzyme	Fragment (bp)
PXEHA3	*Xba* I and *Hind* III	975 + 9020
PXEHA15	*Ecor* I and *Xba* I	3495 + 6887
PXEHA32	*Kpn* I and *Hind* III	3425 + 6962

**Table 6 ijms-26-02979-t006:** Primers required for PCR of the bacterial liquid.

Primers	Fragment (bp)
UvALG3-UF/UvALG3-UR	771
UvALG3-DF/UvALG3-DR	969
UvALG15-UF/UvALG15-UR	1058
UvALG15-DF/UvALG15-DR	1075
UvALG32-UF/UvALG32-UR	1136
UvALG32-DF/UvALG32-DR	1131
Hyg-1/Hyg-2	622

## Data Availability

Data are contained within the article.

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
