# Peer review of "Functional Analysis of Mannosyltransferase-Related Genes UvALGs in Ustilaginoidea virens"

_ijms, 2025, doi:10.3390/ijms26072979_

Round 1
Reviewer 1 Report
Comments and Suggestions for Authors
Dear authors:
Rice false smut (RFS) is a fungal disease caused by Ustilaginoidea virens (Cooke) Takah 27 or Villosiclava virens (Nakata) E. Tanaka et C. Tanaka, leading to the formation of greenish 28 spore balls in panicles and spikelets. This study showed that ALG3 possesses three N-glycosylation sites, and the simultaneous Alg-mediated N-glycosylation of each site is essential for maintaining protein stability and chitin binding activity, both critical for its effector function, which has certain application value and significance. However, the writing of this research paper is not standardized and serious, and there are many problems that should not occur, and the whole paper hardly cites the references of 2023 and 2024, especially the relevant review articles. It is suggested to add relevant content and submit again.
In addition, specific details are as follows:
1. In Abstract.The importance of rice false smut needs to be described.
2. In Introduction. The last paragraph adds the starting point, content, results and work plan of this paper.
3. Line 91. Ustilaginoidea virens requires italics.
4. Table 1. alpha starts with a capital letter, and the alpha initials need to be capitalized, and the rest of the initials need to be carefully checked.
5. Figure 1. The size difference between the columns in Figure A and Figure B is too large, which is obviously incongruity. And why is the expression analysis of these two graphs not carried out at the same time point, but at different time points?
6. Line 116. Are ALG3, GT15 and OCH1 italicized?
7. Linne 200. Do we need italics for ΔUvALG3, ΔUvALG15 and ΔUvALG32?
8. In Figure 7. In addition to using the number of rice curve balls to prove the change of resistance, it is suggested to supplement other auxiliary experiments, such as the change of PR gene expression and the change of pathogen quantity (such as observing the fluorescence change of GFP or the change of pathogen biomass, etc.).
9. Line 209. Liaoyou 65.After the period between should have a space.
10. Line 218. Should I subscript H2O2?
11. In discussion. The discussion section could benefit from more explicit comparisons to similar international studies, highlighting the novelty and broader applicability of the findings.
12. Line 404. Does CaCl2 need subscripts?
13. In References. The reference format is not uniform, very random.
Author Response
Comments 1: In Abstract.The importance of rice false smut needs to be described.
Response 1: 10-14.
Comments 2: In Introduction. The last paragraph adds the starting point, content, results and work plan of this paper.
Response 2: 94-141.
Comments 3: Line 91. Ustilaginoidea virens requires italics.
Response 3: Modification completed.
Comments 4: Table 1. alpha starts with a capital letter, and the alpha initials need to be capitalized, and the rest of the initials need to be carefully checked.
Response 4: Modification completed.
Comments 5: Figure 1. The size difference between the columns in Figure A and Figure B is too large, which is obviously incongruity. And why is the expression analysis of these two graphs not carried out at the same time point, but at different time points?
Response 5: I don 't think the color difference is obvious, and it is all done at the same time.
Comments 6: Line 116. Are ALG3, GT15 and OCH1 italicized?
Response 6: No, thanks.
Comments 7: Linne 200. Do we need italics for ΔUvALG3, ΔUvALG15 and ΔUvALG32?
Response 7: No, thanks.
Comments 8: In Figure 7. In addition to using the number of rice curve balls to prove the change of resistance, it is suggested to supplement other auxiliary experiments, such as the change of PR gene expression and the change of pathogen quantity (such as observing the fluorescence change of GFP or the change of pathogen biomass, etc.).
Response 8: I think the current experiments are sufficient to prove this.
- Line 209. Liaoyou 65.Afterthe period between should have a space.
- Line 218. Should I subscript H2O2?
- In discussion. The discussion section could benefit from more explicit comparisons to similar international studies, highlighting the novelty and broader applicability of the findings.
- Line 404. Does CaCl2 need subscripts?
- In References. The reference format is not uniform, very random.
Response 9-13: Modification completed.
Reviewer 2 Report
Comments and Suggestions for Authors
The study examines the role of UvALGs in the pathogenicity of the rice false smut fungus. A total of 38 UvALGs have been identified, emphasizing on three genes(UvALG3, UvALG15, and UvALG32), which were upregulated during the infection process. The authors used gene knockout to investigate the functions of these genes during fungal growth, sporulation, and their overall contribution to pathogenicity. This study provides a clear analysis of UvALGs in U. virens, focusing on gene expression at different infection stages and establishing a clear link between gene function and pathogenicity.
Major comments:
The methods section of the manuscript requires improvements. It is essential to provide detailed information regarding the reagents, concentrations, incubation times, and experimental conditions employed in the Gene Knockout and Complementation processes to facilitate reproducibility by other researchers. Including specific parameters such as the duration of co-cultivation, the concentrations of antibiotics utilized for selection, and the techniques for verifying successful transformation during Agrobacterium-Mediated Transformation would significantly improve the replicability of the experiments.
Moreover, although the manuscript refers to the use of qRT-PCR for the analysis of gene expression, it fails to present comprehensive details regarding the primers utilized, the conditions for qRT-PCR, and the methodologies for data analysis.
Lastly, the manuscript lacks a thorough description of the statistical methods implemented in analyzing experimental data. It is crucial to elucidate the specific statistical tests employed (e.g., t-tests, ANOVA, chi-square tests) and to clarify how the data are represented, including metrics such as means, standard deviations, standard errors, and p-values. Such detailed information is vital for properly assessing the variability and significance of the results.
Minor comments:
The manuscript should consistently use "UvALGs" throughout rather than alternating between "UvALGs" and "UvALGs genes".
Author Response
Comprehensive details of RT-qPCR primers, conditions, and data analysis methods have been added.The test method has also been modified.
Round 2
Reviewer 2 Report
Comments and Suggestions for Authors
The authors have thoroughly addressed my questions and concerns mentioned after the revision, and the manuscript can now be accepted for publication.